# TRPV4 Mediates Alveolar Epithelial Barrier Integrity and Induces ADAM10-Driven E-Cadherin Shedding

**DOI:** 10.3390/cells13201717

**Published:** 2024-10-17

**Authors:** Lena Schaller, Thomas Gudermann, Alexander Dietrich

**Affiliations:** Walther Straub Institute for Pharmacology and Toxicology, Member of the German Center for Lung Research (DZL), Medical Faculty, LMU-Munich, Nussbaumstrasse 26, 80336 Munich, Germany; lena.schaller@lrz.uni-muenchen.de (L.S.); thomas.gudermann@lrz.uni-muenchen.de (T.G.)

**Keywords:** a disintegrin and metalloprotease 10 (ADAM10), electrical cell–substrate impedance sensing (ECIS), epithelial cadherin (E-cadherin), transient receptor potential vanilloid 4 (TRPV4)

## Abstract

Transient receptor potential vanilloid 4 (TRPV4) channels have been associated with numerous pulmonary pathologies, including hypertension, asthma, and acute lung injury. However, their role in the alveolar epithelium remains unclear. We performed impedance-based resistance measurements in primary differentiated alveolar epithelial type I (AT1) cells from wild-type (WT) and TRPV4-deficient (TRPV4−/−) C57/BL6J mice to detect changes in AT1 barrier integrity upon TRPV4 activation. Both pharmacological (GSK1016790A) and a low pH-driven activation of TRPV4 were quantified, and the downstream effects on adherens junctions were assessed through the Western blotting of epithelial cadherin (E-cadherin) protein levels. Importantly, a drop in pH caused a rapid decrease in AT1 barrier resistance and increased the formation of a ~35 kDa E-cadherin C-terminal fragment, with both effects significantly reduced in TRPV4−/− AT1 cells. Similarly, the pharmacological activation of TRPV4 in AT1 cells triggered an immediate transient loss of barrier resistance and the formation of the same E-cadherin fragment, which was again diminished by TRPV4 deficiency. Moreover, TRPV4-mediated E-cadherin cleavage was significantly reduced by GI254023X, an antagonist of a disintegrin and metalloprotease 10 (ADAM10). Our results confirm the role of TRPV4 in regulating alveolar epithelial barrier permeability and provide insight into a novel signaling pathway by which TRPV4-induced Ca^2+^ influx stimulates metalloprotease-driven ectodomain shedding.

## 1. Introduction

The alveolar epithelial barrier is crucial for maintaining effective gas exchange and protecting the lungs from environmental pathogens and toxicants. This barrier consists primarily of thin alveolar type 1 (AT1) cells, which are responsible for 95% of respiratory gas exchange, and cuboidal alveolar type 2 (AT2) cells, which produce surfactants and act as AT1 progenitor cells [1,2]. Due to its delicate yet restrictive nature, the permeability of the alveolar epithelial barrier is tightly regulated. Disruptions in alveolar barrier integrity can lead to pulmonary edema, characterized by the accumulation of protein-rich extravascular fluid in the interstitium and alveoli. Extensive pulmonary edema can develop into acute respiratory distress syndrome (ARDS), with an approximate mortality rate of 30–40% [3].

The integrity of the alveolar epithelium strongly depends on the intercellular junctions that regulate paracellular permeability. These junctions, including tight junctions (TJs), adherens junctions (AJs) and desmosomes, also facilitate cell-cell communication and maintain cell polarity [4]. While TJs are primarily responsible for maintaining a restrictive paracellular barrier, AJs play integral roles in the formation and regulation of TJs and their associated proteins [4,5,6,7]. Epithelial AJs consist of Ca^2+^-dependent homotypic adhesions between the extracellular regions of E-cadherin proteins on neighboring cells [4]. The cytoplasmic tail of E-cadherin, like other classical cadherins, is highly conserved and interacts with the actin cytoskeleton through the anchor proteins p120 catenin, β-catenin, and α-catenin [4,6,8]. The absence of E-cadherin has been associated with hallmarks of altered barrier integrity, including increased cell proliferation, motility, and invasiveness [5,9,10,11,12].

The E-cadherin proteins of AJs undergo constant turnover and are readily ubiquitinated, endocytosed, and returned to the plasma membrane, possibly through a protein kinase C- and/or Rho-dependent pathway [9,13,14]. However, these proteins are also susceptible to cleavage by various proteases, resulting in multiple cleavage products that may possibly trigger downstream signaling cascades. For instance, the intracellular C-terminal region of E-cadherin can be cleaved by gamma-secretase and caspase, while the extracellular region is targeted by numerous metalloproteases [8,15,16,17]. One such metalloprotease, a disintegrin and metalloprotease 10 (ADAM10), is widely expressed in epithelial tissue and plays a significant role in cadherin cleavage [15,18]. ADAM10 activation requires intracellular Ca^2+^ influx, although the exact upstream pathway leading to ADAM10 activation remains unclear [19].

Recent research suggests that members of the transient receptor potential (TRP) superfamily may initiate the Ca^2+^ influx necessary for ADAM10 activation [19]. Of the potential candidates, the fourth member of the vanilloid family (TRPV4) is of particular interest. TRPV4, like other TRP proteins, has intracellular N- and C-termini and consists of six transmembrane domains with a pore-forming loop spanning helices 5 and 6 [20,21,22]. TRPV proteins are identified through their long N-terminal ankyrin repeat domains and typically form homotetrameric nonselective ion channels [20,23]. TRPV4 channels are expressed in many organ systems, including the lungs, where they are found in immune cells, endothelial cells, and epithelial cells of the trachea, bronchi, and alveoli [21,22,24,25,26,27]. Functional TRPV4 channel homotetramers are mechano-, pH-, osmo- and thermosensitive, and are also activated by chemical mediators including phorbol esters and arachidonic acid metabolites [22,24,28].

The roles of TRPV4 channels in pulmonary injury are complex. On one hand, TRPV4 ablation increased pulmonary edema formation in a model of ischemia-reperfusion, highlighting an important function in the chronic expression and regulation of proteins for the protection of cell barrier integrity ([27] reviewed in [22]). On the other hand, isolated lungs from TRPV4−/− mice developed significantly reduced edema following ventilation-induced lung damage, most probably due to the absence of these channels in endothelial cells and their acute activation by mechanical stress [29]. While TRPV4’s role in inducing pulmonary endothelial barrier permeability is well documented (reviewed in [20,21,22,24,25,30,31]), its acute effects on the alveolar epithelial barrier are less understood. Studies on epithelial cell lines have shown conflicting results. The activation of TRPV4 induced barrier permeability and altered tight junction morphology in a mouse mammary cell line HC11 [32] and Madin-Darby canine kidney II [33] monolayers. Conversely, in corneal epithelial RCE1(5T5) cells [34] and keratinocytes [35], TRPV4 activation increased barrier resistance and upregulated the expression of TJ-associated proteins. Isolated alveolar epithelial cells from rats exhibited decreased barrier resistance immediately after exposure to a specific TRPV4 activator [36]. Observations of TRPV4-mediated blebbing in endothelial and epithelial cells in the alveoli [29] demand further investigation of the molecular mechanisms of the AT1 cell-induced loss of cell barrier function by TRPV4-driven Ca^2+^ entry.

In addition to its association with fibrosis and pulmonary hypertension (reviewed in [24,25]), TRPV4 has also been linked to acid-induced acute lung injury (ALI) [30,37]. Acid-induced ALI can occur in cases of occupational exposure, as well as in patients suffering from gastroesophageal reflux disease (GERD), the latter of which is a known risk factor for recurrent ALI [38]. Two independent studies in 2014 [30] and 2016 [37] working with similar murine models of acid-induced ALI found that TRPV4−/− mice were protected from inflammation and pulmonary edema following intratracheal HCl instillation, with reduced inflammatory cytokine levels and neutrophil recruitment in the lungs. Acid-induced ALI may be in part due to the activity of TRPV4 in immune cells. In 2010, Hamanaka et al. showed that TRPV4-expressing macrophages could restore the susceptibility of TRPV4−/− lungs to mechanically induced lung injury [26]. However, transcriptomic analysis identified only a minimal TRPV4 expression in neutrophils [30], suggesting TRPV4 may influence neutrophil function and recruitment indirectly, possibly by controlling AT1 paracellular permeability.

Only a limited number of studies indicate that TRPV4 activation may induce instability in the alveolar epithelial barrier, similar to its effect on microvascular endothelial cells. To validate these findings and further explore the underlying mechanisms, we isolated AT1 cells from WT and TRPV4−/− mice and assessed the effects of pH- and agonist-driven TRPV4 activation on alveolar epithelial barrier integrity through electrical cell–substrate impedance sensing (ECIS) [39,40]. We determined that TRPV4 activation triggered an immediate but transient reduction in barrier resistance, accompanied by the ADAM10-mediated cleavage of E-cadherin.

## 2. Materials and Methods

### 2.1. Animals

TRPV4−/− (B6.199X1-Trpv4^tm1MSZ^ from Riken BioResource Research Center RBRC01939, Ibaraki, Japan) mice were backcrossed 10 times with the C57/BL6J strain. The correct knockout of the TRPV4 protein was approved by Western blotting in our recent manuscript [27]. Sex- and age-matched mice between 2 and 4 months of age and wild-type controls from the same colony were used in all experiments.

### 2.2. Isolation and Culture of Primary Alveolar Epithelial Cells and In Vitro Differentiation of AT1 Cells

The isolation of primary alveolar epithelial cells from murine lungs was performed as previously described [27]. In brief, 3–6 mice were sacrificed via cervical dislocation. Lungs were transcardially perfused with 20 mL of Dulbecco’s phosphate-buffered saline (PBS, Merck, Darmstadt, Germany, D8537), inflated intratracheally with 1.5 mL of dispase solution (Corning, New York, NY, USA, 354235), followed by 400 µL of 1% low gelling temperature agarose (Merck, A9414) in Dulbecco’s Modified Eagle Medium (DMEM, Thermo Fisher Scientific, Waltham, MA, USA, 41965039). Once the agarose had solidified, lungs were resected and digested in a 1 mL dispase solution for 45 min at room temperature (RT). Lung lobes were then manually dissociated in 5 mL HEPES-buffered DMEM with 100 U/mL DNase I (AppliChem, Darmstadt, Germany, A3778). The tissue suspension from each mouse was pooled and passed through a series of 100 µm, 20 µm, and 10 µm filters (Sefar, Helden, Switzerland, 3A03-0010-102-00, 3A03-0020-102-10 and 3A03-0100-115-01) to ensure a dispersed cell suspension. The resulting suspension was then centrifuged (10 min, 200× *g*), the media removed, and the pellet resuspended in HEPES-buffered DMEM. The cell suspension was then plated out on CD16/32- and CD45-coated (BD Biosciences, Franklin Lakes, NJ, USA, 553142 and 553076) Petri dishes in a negative selection step for macrophages and lymphocytes. Following a 30 min incubation step at 37 °C, the dishes were washed thrice with HEPES-buffered DMEM, and the nonadherent cells were transferred to uncoated, tissue culture-treated 10 cm dishes (Sarstedt, Nümbrecht, Germany, 83.3900.300). After allowing 1 h for fibroblast deposition, the suspension was carefully removed from all plates and centrifuged (see above settings). The media was aspirated, and the resulting AT2 cell pellet was resuspended in culture media (HEPES-buffered DMEM supplemented with 10% FCS, 1% P/S) and seeded according to intended experimental requirements. Cells were either harvested after 48 h in culture for AT2 immunocytochemistry or allowed to differentiate to AT1 cells over 7 days.

### 2.3. Indirect Immunocytochemistry

Isolated WT alveolar epithelial cells were seeded on poly-L-lysine-coated 12 mm glass coverslips. After 48 h or 7 days of culture at 37 °C and 5% CO_2_, cells were washed once with cold PBS, fixed in 4% PFA/PBS (15 min, RT), and then washed thrice with cold PBS. Cells were permeabilized for 10 min at RT in a 0.2% Triton X-100/PBS solution and then washed 4 × 5 min in PBS-T (0.1% Tween20 in PBS). Cells were blocked for 1 h in PBS containing 0.1% Tween20 and 5% BSA, after which they were incubated overnight at 4 °C in primary antibody solutions prepared in blocking buffer (see Appendix A for antibodies and dilutions). The following day, cells were washed (4 × 5 min, PBS-T), incubated for 2 h at RT in secondary antibody solutions, and subsequently washed (4 × 5 min, PBS-T). All antibodies were diluted in the blocking buffer. Next, nuclei were stained with DAPI (0.1 mg/L in PBS) for 3 min at RT, after which cells were washed (4 × 5 min, PBS-T). Coverslips were mounted with PermaFluor mounting medium (Epredia, Kalamazoo, MI, USA, #TA-030-FM) and kept at 4 °C. Confocal images were taken with a Zeiss LSM 880 microscope using the ZEN Black software (2.3 SP1 FP3). Images were processed with FIJI software (Image J v.1.53c, Wayne Rasband, NIH, Bethesda, MD, USA) [41].

### 2.4. Ca^2+^ Imaging

Isolated TRPV4−/− and WT AT2 cells were grown on 24 mm glass coverslips for 7 days. On the day of measurement, differentiated AT1 cells were loaded with 2 µM Fura-2-AM (Merck, #47989-1MG-F) in Ca^2+^ buffer (0.1% BSA in HBSS (with Ca^2+^, Mg^2+^ and 0.5 M HEPES)) for 25 min at RT. Coverslips were then washed with HEPES/HBSS buffer, inserted in a quick-change chamber (Warner instruments, Holliston, MA, USA, #64-0367) with 450 µL HEPES/HBSS, and placed on the 40x oil objective of a Leica DM98 fluorescence microscope. Any changes in intracellular Ca^2+^ concentration following TRPV4 activation (100 nM GSK1016790A, GlaxoSmithKline, Brentford, UK) were recorded at 340 and 380 nm wavelengths, as described [42].

### 2.5. Quantification of Alveolar Epithelial Barrier Resistance

Freshly isolated WT and TRPV4−/− AT2 cells were seeded at a density of 2 × 10^4^ cells/well on electrical cell–substrate impedance sensing (ECIS) plates (Applied Biophysics, Troy, NY, USA, 8W10E+), which had been treated with 10 mM of L-Cysteine, as per the manufacturer’s recommendation. Cells were kept in culture at 37 °C and 5% CO_2_ for 7 days, at which point barrier integrity experiments were conducted, with resistance measured at 500 Hz using the ECIS ZΦ device (Applied Biophysics).

### 2.6. SDS-PAGE and Western Blot Analysis

The expression of full-length and degraded E-cadherin protein were evaluated by Western blot analysis, as previously described [42]. Following treatment, AT1 cells were lysed in 100 µL of RIPA buffer (20 mM Tris-HCl, pH 7.5, 150 mM NaCl, 1% Nonidet P40, 0.5% sodium deoxycholate, 1% SDS, 5 mM EDTA) with protease and phosphatase inhibitors (Roche, Mannheim, Germany, #04906837001, #05892791001) for 30 min on ice and sonicated for 30 s. Protein concentration was quantified with the Pierce BCA Protein Assay Kit (Thermo Fisher Scientific, #23225) according to the manufacturer’s protocol. Prepared protein samples (5 µg lysate, 1× Laemmli buffer (prepared from 5× stock: 3 mL TRIS/HCl (2.6 M), pH 6.8; 10 mL glycerin; 2 g SDS; 2 mg bromophenol blue; 5 mL β-mercaptoethanol)) were heated for 10 min at 95 °C, then loaded onto an SDS-PAGE gel (4% stacking, 10% separating). Gel electrophoresis was run for 30 min at 80 V, then at 120 V for 90 min. Proteins were then transferred from the gel to a Roti^®^-PVDF membrane (Roth, Karlsruhe, Germany, #T830.1) in a wet transfer system (BioRad, Feldkirchen, Germany) at 50–60 V for 1.5 h. After the transfer, the membrane was blocked with 5% low-fat milk (Roth, #T145.2) in TBS-T (0.1% Tween20) for 1 h at RT. All antibodies were diluted in the milk blocking solution. Membranes were incubated in the primary antibody solutions overnight at 4 °C. The next day, membranes were washed (3 × 10 min, TBS-T) and incubated for 2 h at RT in peroxidase-conjugated secondary antibody solutions. Chemiluminescence was imaged following incubation in SuperSignal West Femto or Pico maximum sensitivity substrates (Life Technologies, Carlsbad, CA, USA, #34095 and #34580), using an Odyssey Fc unit (Licor, Lincoln, NE, USA). For antibody information and dilutions, see Appendix A.

### 2.7. Statistical Analysis

Statistical tests were performed using GraphPad Prism 10 software (GraphPad Software, San Diego, CA, USA). Significant differences are indicated by asterisks, where *p* < 0.05 (*), 0.01 (**), 0.001 (***), and 0.0001 (****).

## 3. Results

### 3.1. Differentiation and Characterization of Primary Murine Alveolar Epithelial Type I Cells

In order to assess the effects of TRPV4 activation upon AT1 cells, we first validated our isolation and differentiation procedure. Freshly isolated murine AT2 cells were fixed 2 and 7 days after isolation and stained for the epithelial markers prosurfactant protein C (pSPC) and aquaporin-5 (AQP5), which are specifically expressed by AT2 and AT1 cells, respectively (Figure 1) [43]. The process of AT2-to-AT1 differentiation was clearly delineated, with the 2-day epithelial cells staining positive for pSPC and negative for AQP5, and the 7-day epithelial cells only showing positive staining for AQP5. These results confirm that our isolation protocol yields a small population of AT2 cells, which differentiate into a confluent monolayer of AT1 cells within 7 days, as previously described [27].

### 3.2. TRPV4 Mediates Acid-Induced Alveolar Epithelial Barrier Dysfunction

As TRPV4 is known to be activated under acidic conditions, we tested the effect of HCl application on isolated AT1 cells to represent acid-induced ALI. We obtained real-time quantitative measurements of AT1 barrier integrity through electrical cell-substrate impedance sensing (ECIS). A drop in media pH from 7.5 to 4.5 induced a rapid decrease in AT1 barrier resistance in WT cells (Figure 2A). Although TRPV4−/− AT1 cells also experienced a drop in resistance, it was less pronounced than in WT AT1 cells. The difference between genotypes became apparent as soon as 10 min after exposure, with a barrier resistance of only 12% ± 3.5% in WT AT1 cells, which was significantly decreased compared to TRPV4−/− cells (Figure 2B). Along this line, the drop in pH had a strong impact on the AT1 monolayer and affected cell–matrix adhesion, as well as cell–cell junctions, as indicated by changes in the monolayer capacitance. AT1 monolayers of both genotypes showed a noticeable increase in capacitance upon media acidification, although the increase was only significant in WT cells (Appendix A). At the protein level, HCl exposure resulted in an increased formation of a ~35 kDa C-terminal fragment (CTF) of E-cadherin (Figure 2C). The amount of this CTF was increased in all HCl-treated samples but was more pronounced in lysates from WT AT1 cells. Quantification revealed that the amount of E-cadherin CTF generated in WT HCl-treated cells was significantly larger than in the respective control samples and HCl-treated TRPV4−/− cells (Figure 2D). HCl exposure did not significantly alter the levels of full-length E-cadherin in any genotype (Appendix A).

### 3.3. Pharmacological Activation of TRPV4 Induces a Rapid Transient Drop in Barrier Resistance

To better explore the mechanisms underlying TRPV4-mediated AT1 barrier dysfunction, we applied characterized pharmacological modulators of TRPV4. We tested a specific TRPV4 activator, GSK1016790A, with an EC_50_ of 5 nM (GSK101, Tocris, #6433 [44]), on differentiated AT1 cells using Ca^2+^ imaging, along with the TRPV4 inhibitor GSK2193874, with an IC_50_ of 2–40 nM (GSK219, Tocris, #5106 [45]). The application of GSK101 in WT AT1 cells resulted in a transient increase in intracellular Ca^2+^ ([Ca^2+^]_i_), which was absent in WT AT1 cells pre- and co-treated with GSK219 (Figure 3A). In TRPV4−/− AT1 cells, the application of GSK101 had no effect on [Ca^2+^]_i_ (Figure 3B).

With the specificity of GSK101 in AT1 cells confirmed, we next examined the effect of TRPV4 activation upon AT1 barrier integrity. Dose–response ECIS experiments established that 100 nM GSK101 was the optimal concentration for TRPV4-driven AT1 barrier disruption, as the effect plateaued at higher concentrations (up to 3 μM, Appendix A). The activation of TRPV4 resulted in a transient drop in barrier resistance in WT AT1 cells, peaking approximately 15 min after GSK101 application and recovering to baseline levels after 90 min (Figure 3C). In line with our previous results on TRPV4-induced [Ca^2+^]_i_, GSK101 had no effect on AT1 cell permeability after pre- and co-treatment with GSK219, nor in TRPV4−/− cells. The quantification of the barrier function 15 min after GSK101 exposure revealed that the 13% ± 2.7% loss of cell resistance in WT AT1 cells was significantly larger than in untreated controls and GSK219-treated WT cells (Figure 3D). Similarly, the loss in barrier integrity upon GSK101 exposure in WT AT1 cells was significantly larger than in TRPV4−/− AT1 cells (Appendix A). In all ECIS experiments, the capacitance of the AT1 monolayer remained constant, indicating that the observed changes in resistance were due to altered cell–cell junction integrity and not cell-substrate detachment (Appendix A).

### 3.4. TRPV4 Activation Triggers an ADAM10-Mediated Cleavage of E-Cadherin

The drop in AT1 cell barrier resistance following TRPV4 activation suggested a loss in paracellular junction integrity. One possible explanation for this sudden drop in resistance might be a TRPV4-driven activation of one or more metalloproteinases, causing a rapid shedding of the AT1 protein ectodomain [19]. Through a Western blot protein analysis, we identified an increased presence of a ~35 kDa E-cadherin CTF 15 min after treatment with GSK101 (Figure 4A). This CTF was detected in far lower quantities in samples pre- and co-treated with GSK219 and was absent in TRPV4−/− AT1 cells. As E-cadherin is a known substrate of ADAM10 [18], we also assessed whether treatment with an ADAM10 inhibitor, GI254023X, with an IC_50_ of 5.3 nM (GI254, Tocris, #3995 [46]), would limit the TRPV4-driven formation of this CTF. Western blot quantification confirmed that the significant increase in the E-cadherin CTF formation upon GSK101 application in WT AT1 cells was entirely dependent on TRPV4, as pre- and co-incubation with GSK219 maintained E-cadherin CTF levels equal to those in untreated controls (Figure 4A,B). Additionally, ADAM10 inhibition through GI254 partially but significantly reduced the GSK101-driven formation of the E-cadherin CTF (Figure 4A,B). In TRPV4−/− AT1 cells, Western blot quantification revealed no significant change in E-cadherin CTF formation after GSK101 exposure (Appendix A). Additionally, in all treatment groups and genotypes, the effect of TRPV4 activation on the expression of full-length E-cadherin protein levels was not significantly different (Appendix A).

## 4. Discussion

TRPV4 is a well-recognized mediator of lung function and has also been implicated in various pulmonary disease states, including fibrosis, inflammation, and pulmonary edema formation [22,24,25,27,47]. Numerous studies in endothelial cells and pulmonary arterial smooth muscle cells have demonstrated that TRPV4 activation increases monolayer permeability through mechanisms such as the downregulation of TJ-associated genes, the contraction of actin–myosin rings, the disorganization of F-actin, and the loss of cell–matrix adhesion [20,25,32,48,49]. However, the role of TRPV4 in alveolar epithelial barrier function and integrity is still elusive. Through real-time impedance measurements and protein analysis of isolated differentiated murine AT1 cells, we confirmed that TRPV4 activation induces a rapid transient drop in AT1 barrier resistance. Here, we showed for the first time that TRPV4 activation in primary AT1 cells destabilizes AJs through an ADAM10-mediated extracellular cleavage of E-cadherin. Additionally, we found that TRPV4 mediates AT1 barrier dysfunction in a model of acid-induced ALI, suggesting that TRPV4-driven disruption of paracellular barrier integrity might occur, irrespective of the initial stimulus.

We began by assessing the role of TRPV4 in a clinically relevant context. Patch clamp electrophysiology performed in Chinese hamster ovary (CHO) cells demonstrated that TRPV4 responds significantly to low pH [28]. As the pH necessary to open the channel is lower than physiological levels in most compartments, the pH activation of TRPV4 is mainly relevant in conditions of acid-induced ALI, as occurs through occupational exposure or in patients suffering from gastroesophageal reflux disease (GERD) [50]. Consistent with in vivo results from previous studies [30,37], we observed that TRPV4 deficiency reduces the effects of low pH on AT1 barrier integrity. Both the HCl-induced drop in barrier resistance and the accompanying degradation of E-cadherin were significantly increased in WT AT1 cells compared to those from TRPV4−/− mice. Thus, by facilitating barrier permeabilization, TRPV4 in alveolar epithelial cells may also support the neutrophil recruitment observed in whole lungs following HCl instillation [30,37].

Our ECIS experiments in WT AT1 cells revealed that unlike HCl exposure, the drop in barrier resistance following the application of the specific TRPV4 activator GSK101 was transient, with a return to baseline resistance within 90 min of exposure. This discrepancy may be due to the caustic effects of the acidic media. Studies in the human alveolar cell line A549 have demonstrated that incubation with low-pH media suppresses cell proliferation rates [51] and induces significant persistent reductions in barrier resistance [52]. Few studies have assessed GSK101-driven changes in epithelial barrier resistance, showing varying results. Martinez-Rendon et al., in 2016, found that GSK101 treatment led to a gradual increase in the transepithelial resistance (TER) of corneal epithelial cells from the RCE1(5T5) cell line [34]. However, these results reflect measurements taken hours after GSK101 application, while we and others have shown that dynamic changes in barrier resistance occur within the first 30 min of TRPV4 activation in primary differentiated AT1 cells [36,53].

While the inhibitory effects of GSK219 pretreatment on GSK101-induced TRPV4 activity (Figure 3A), the loss of AT1 barrier resistance (Figure 3C,D), and E-cadherin degradation (Figure 4) were very effective, we are also aware that two other TRPV4 inhibitors applied after HCl incorporation that were able to suppress acute lung injury in vivo [30]. Whether GSK219 is similarly effective in post-exposure treatments needs to be explored.

Our results also give some insight into the downstream mechanisms underlying TRPV4-induced AT1 barrier dysfunction. As with our model of acid-induced ALI, GSK101 application increased the formation of a ~35 kDa C-terminal fragment of E-cadherin, detected as early as 15 min after exposure, at the point of greatest loss of barrier resistance. The formation of this fragment was TRPV4 specific, as its levels did not increase in TRPV4−/− AT1 cells or cells treated with the specific TRPV4 antagonist GSK219. The size of this fragment indicates an extracellular metalloprotease cleavage event. A recent study by Tatsumi et al. proposed that TRP-induced Ca^2+^ influx could activate certain ADAM proteins, including ADAM10 and ADAM17, by the Ca^2+^ sensitive protein ANO6 [54], leading to the ectodomain shedding of their respective ligands [19]. Although mechanistic connections between TRPV4 and the matrix metalloproteases MMP2 and MMP9 have been reported [55], a physiological relationship between TRPV4 channels and ADAM10 had not yet been identified [19]. Using an ADAM10-specific antagonist, we showed that TRPV4-driven E-cadherin cleavage in AT1 cells is partially mediated by ADAM10 (Figure 5). As it is likely that additional MMPs and ADAM proteins are activated by TRPV4-induced Ca^2+^ influx, future experiments characterizing the resulting ectodomain cleavage would be beneficial to determine to what extent TRPV4-induced cell detachment is dependent on protease activity.

Altered alveolar epithelial barrier integrity can often lead to the development of pulmonary edema, which can progress to acute respiratory distress syndrome (ARDS) Matthay, Zemans, Zimmerman, Arabi, Beitler, Mercat, Herridge, Randolph and Calfee [3]. It is possible, however, that increased barrier permeability could be physiologically beneficial. Edema fluid could dilute inflammatory mediators, neutralize an acidic environment, or carry pathogen-combating neutrophils. Along this line, there is strong biological support for a mechanism by which TRPV4 at the plasma membrane of AT1 cells facilitates neutrophil transmigration. E-cadherin and TRPV4 show similar staining patterns at the plasma membrane in various epithelial cell types [32], with immunoprecipitation pulldown experiments revealing a molecular connection between TRPV4 and α-catenin, β-catenin, and E-cadherin [56]. Epithelial TRPV4 is localized to the basolateral membrane, which comprises the alveolar septal wall [29,32]. Neutrophil adhesion to the alveolar epithelium is restricted to the basolateral side, with eventual transmigration across the epithelial barrier limited to the paracellular route [57]. TRPV4’s localization to the basolateral membrane, paired with its colocalization with E-cadherin, would allow targeted Ca^2+^ influx at sites requiring AJ weakening. The transient drop in AT1 barrier resistance upon GSK101 application indicates that the barrier disruption induced by TRPV4 activity is quickly resolved. Therefore, the TRPV4-ADAM10-E-cadherin cleavage may facilitate efficient regulated neutrophil transmigration.

In addition to the alveolar epithelium, the alveolar barrier also includes the microvascular endothelium. Our ECIS measurements in WT AT1 cells are consistent with those previously performed in human lung microvascular endothelial cells, characterized by a ~10–20% drop in barrier resistance within 15 min of GSK101 application and a return to baseline resistance within 3 h [53]. Although a pathway involving TRPV4 and vascular endothelial cadherin (VE-cadherin) cleavage has not yet been investigated, it is possible that metalloprotease activity occurs downstream of TRPV4. Activation occurs in endothelial cells as well, further supporting neutrophil transmigration from the microvasculature to the alveolar epithelium.

## 5. Conclusions

In summary, our study highlights the significant role of TRPV4 in regulating alveolar epithelial barrier integrity. We confirmed that TRPV4 is involved in acid-induced lung injury, as channel activation under conditions of low pH triggered an immediate drop in AT1 barrier resistance with a destabilization of AJ proteins. These observations were corroborated using specific agonists and antagonists of TRPV4. In addition, we discovered a novel mechanism by which TRPV4 activation affects AT1 cell junctions, namely through the ADAM10-mediated cleavage of E-cadherin. These insights into the TRPV4-ADAM10-E-cadherin pathway may be confirmed in an ex vivo lung model in the future [58] and provide a basis for further research into targeted therapies for pulmonary diseases involving epithelial barrier dysfunction.

## Figures and Tables

**Figure 1 cells-13-01717-f001:**
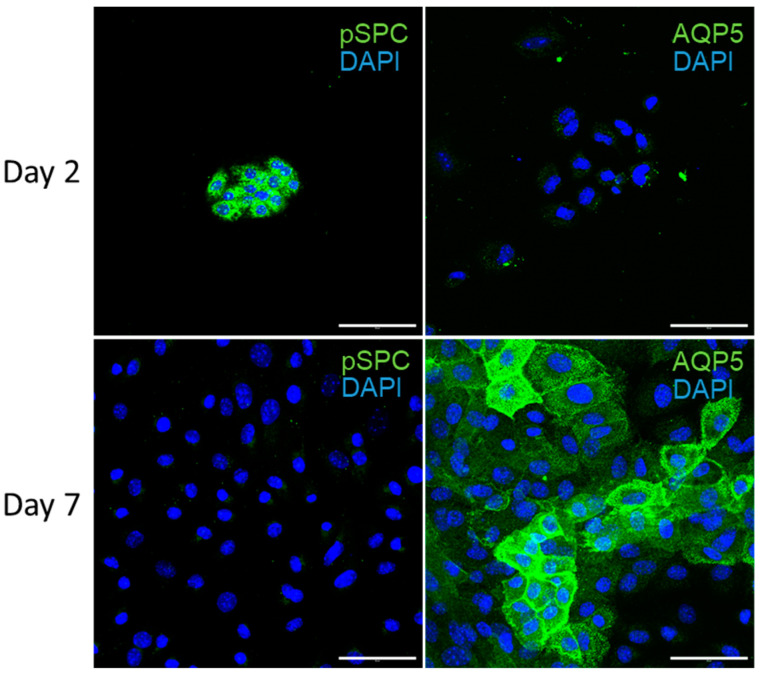
Differentiation and characterization of isolated primary murine alveolar epithelial type I (AT1) cells. Alveolar epithelial cells were fixed 2 (day 2) and 7 days (day 7) after isolation and stained for the AT2 and AT1 markers prosurfactant protein C (pSPC) and aquaporin 5 (AQP5), respectively. Nuclei staining was performed with DAPI dye (DAPI). Scale bar: 50 μm.

**Figure 2 cells-13-01717-f002:**
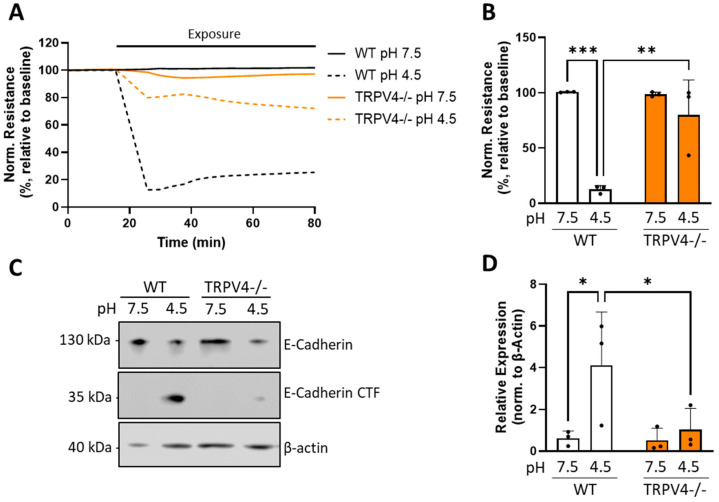
Changes in normalized electrical cell resistance (**A**,**B**) and expression/proteolyis of E-cadherin (**C**,**D**) of wild-type (WT) and TRPV4−/− AT1 cells. Cell resistance was recorded at 500 Hz for 1 h following an HCl-induced drop in media pH (**A**) and normalized monolayer resistance values 10 min after exposure were quantified (**B**). Cells from the same isolations and treatment conditions were lysed after 1 h of exposure, levels of E-cadherin and E-cadherin CTF were assessed by Western blotting (**C**) (see original blot in Appendix A), and the results of the latter were summarized (**D**). Data are presented as mean ± SD (**B**,**D**) from 3 independent cell isolations of 6 mice each (n = 3). Significance between means was analyzed using a two-way ANOVA; * *p* < 0.05, ** *p* < 0.01, *** *p* < 0.001.

**Figure 3 cells-13-01717-f003:**
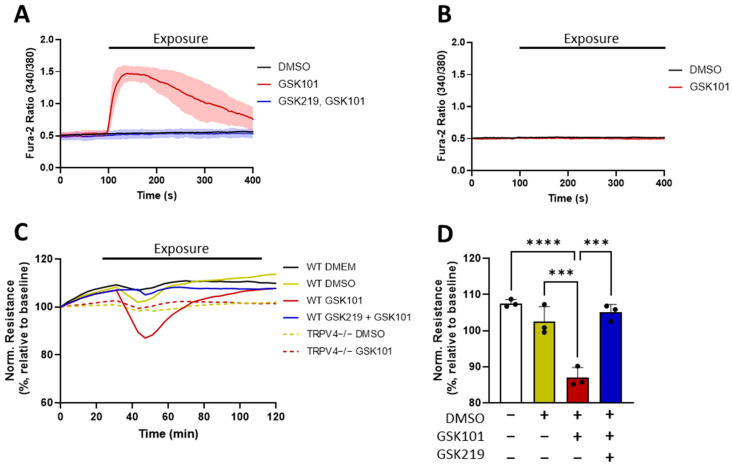
Intracellular Ca^2+^ concentration ([Ca^2+^]_i_) (**A**,**B**) and cell barrier function (**C**,**D**) of AT1 cells after activation (GSK1016790A (GSK101)) and inhibition (GSK2193874 (GSK219)) of TRPV4 channels. [Ca^2+^]_i_ was quantified in primary differentiated AT1 cells from WT (**A**) and TRPV4−/− mice (**B**) upon application of a specific TRPV4 activator (GSK101, 100 nM) in the presence and absence of a specific TRPV4 blocker (GSK219, 300 nM). One representative experiment (n = 10 cells, mean ± SD) out of three independent isolations is shown. Changes in electrical cell resistance (normalized to baseline levels) were recorded with an ECIS device at 500 Hz for WT and TRPV4−/− AT1 cells upon application of GSK101 (100 nM) in the presence and absence of GSK219 (300 nM) for 90 min (**C**). Data represent mean values from 3 independent cell isolations of 6 mice each (n = 3). The normalized electrical cell resistance for WT treatment groups 15 min after exposure was quantified (**D**). Data represent mean ± SD, and significance between means was analyzed with a one-way ANOVA; *** *p* < 0.001, **** *p* < 0.0001.

**Figure 4 cells-13-01717-f004:**
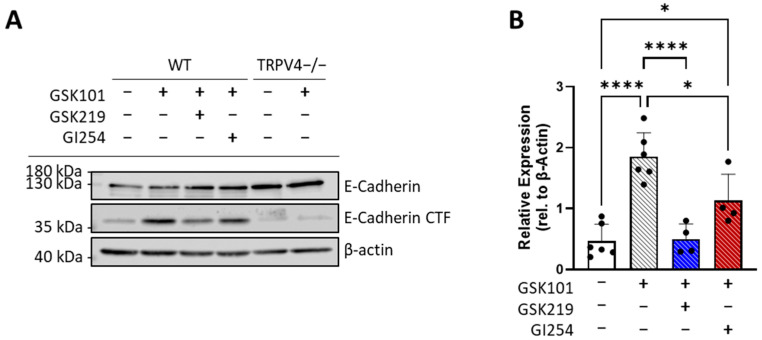
Quantification of E-cadherin and E-cadherin C-terminal fragment (CTF) by Western blotting of protein lysates from primary differentiated AT1 cells isolated from wild-type (WT) and TRPV4-deficient (TRPV4−/−) mice (**A**,**B**). Representative Western blot showing levels of E-cadherin and E-cadherin CTF in WT and TRPV4−/− AT1 cells 15 min after application of GSK101 (100 nM) in the presence and absence of either GSK219 (300 nM) or an ADAM10 inhibitor GI254023X (GI254 3 μM) (**A**) (see original blot in Appendix A). Changes in the levels of E-cadherin CTF in WT AT1 cells were quantified 15 min after exposure to the indicated compounds (**B**). Data represent mean ± SD (**B**) from at least 4 independent cell isolations from 3 to 5 mice, each (n = 4–6). Significance between means was analyzed with one-way ANOVA; * *p* < 0.05, **** *p* < 0.0001.

**Figure 5 cells-13-01717-f005:**
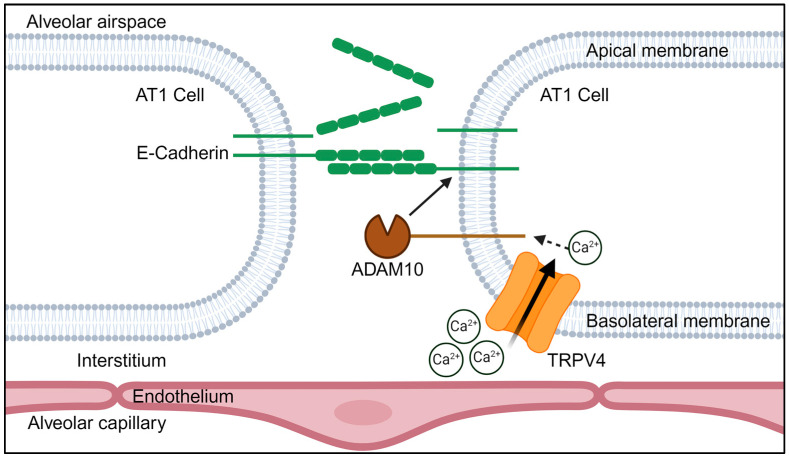
Schematic describing a possible interaction between TRPV4 and ADAM10, resulting in ectodomain shedding of E-cadherin in AT1 cells (modified from [19]). See text for more details.

## Data Availability

All data supporting the findings of this study are available in the paper and its Appendix A section.

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
