# Peer review of "TRPV4 Mediates Alveolar Epithelial Barrier Integrity and Induces ADAM10-Driven E-Cadherin Shedding"

_cells, 2024, doi:10.3390/cells13201717_

Round 1

Reviewer 1 Report

Comments and Suggestions for Authors

The manuscript by Schaller, L. et al. studying the role of TRPV4 in alveolar cells depicts an interesting and convincing experiment. The style and extension are fine. The language is generally fine. I liked the manuscript.

There are some minor comments:

- Materials, page 3, lines 118-122. This section should be more informative and have more information. Riken BioResource Center should have a city and a country. How many animals were used? According to the abstract, the animals are TRPV4 knockouts and are named C57/BL6J strain, some (brief) info about this would be appreciated. You mention nothing about wild type animals…

- Materials, page 3, line 127. Although the word euthanasia is (sadly) widely used in the scientific literature, it is not applicable to the applied procedure. Laboratory animals are commonly killed, or sacrified if you prefer. You should check a dictionary.

- Materials and methods, page 4, lines 153-167. The mentioned procedure has no mention about the use of antibodies. I don´t know if you performed direct or indirect inmmunocytochemistry. May be you should mention here the presence of a supplemental table about the antibodies.

- References to “n” numbers are few along the manuscript…

- Results, page 5, line 221. The specific expression profiles of ATI and ATII cells should have a citation.

- Results, page 5, figure 1. I think it would be clearer if you pair the 4 images by the time they were taken (Day 2: 2 images; Day 7: 2 images).

Author Response

Manuscript Cells-3248583

Response to Reviewers

We thank both reviewers for their helpful comments. All changes according to the reviewers’ comments are marked in red in the revised manuscript.

Reviewer 1

Reviewer comment: Materials, page 3, lines 118-122. This section should be more informative and have more information. Riken BioResource Center should have a city and a country. How many animals were used? According to the abstract, the animals are TRPV4 knockouts and are named C57/BL6J strain, some (brief) info about this would be appreciated. You mention nothing about wild type animals…

Author response: As recommended by the reviewer, we have expanded the materials and methods for section 2.1 Animals with additional information for the Riken BioResource Research Center. The number of mice used for cell isolation is detailed in page 3, line 127. Additionally, information regarding the TRPV4 knockout mice and their background strain of C57/BL6J is stated in page 3, lines 118-120. We also mention the use of age-matched, wild-type control mice in lines 121-122.

Reviewer comment: Materials, page 3, line 127. Although the word euthanasia is (sadly) widely used in the scientific literature, it is not applicable to the applied procedure. Laboratory animals are commonly killed, or sacrified if you prefer. You should check a dictionary.

Author response: We thank the reviewer for bringing attention to the word choice for the killing of animals for scientific research. For clarity and accuracy, we changed the word “euthanized” in page 3, line 127 to “sacrificed”.

Reviewer comment: Materials and methods, page 4, lines 153-167. The mentioned procedure has no mention about the use of antibodies. I don´t know if you performed direct or indirect inmmunocytochemistry. May be you should mention here the presence of a supplemental table about the antibodies.

Author response: We agree with the reviewer’s emphasis on clear presentation of the antibodies used in indirect immunocytochemistry. Information about the method and all the relevant antibodies is provided on page 4 line 152 and in supplementary table S1, as stated in page 4, line 159.

Reviewer comment: References to “n” numbers are few along the manuscript…

Author response: We appreciate the reviewer’s suggestion to add detail regarding the “n” numbers throughout the manuscript. We have added the “n” value for the relevant experiments in the figure legends of Figure 2 (page 6, line 263-264), Figure 3 (page 8, line 302), and Figure 4 (page 8, line 336).

Reviewer comment: Results, page 5, line 221. The specific expression profiles of ATI and ATII cells should have a citation.

Author response: The reviewer makes a good point that the stated ATI and ATII cell markers require a citation. We have amended the manuscript to include this (page 5, line 222; and reference #44).

Reviewer comment: Results, page 5, figure 1. I think it would be clearer if you pair the 4 images by the time they were taken (Day 2: 2 images; Day 7: 2 images).

Author response: We agree with the reviewer’s note that the information in Figure 1 (page 5) would be better conveyed if the images were organized by day, and have reformatted the figure accordingly.

Reviewer 2 Report

Comments and Suggestions for Authors

This is a well conducted and well written study on the role of TRPV4 in epithelial integrity of the alveoli and potential underlying mechanisms. The study is of high relevance and provides a new mechanism for previously observed effects of the activation of TRPV4. 

Minor concerns

1. The AT1 might be more instructive than ATI.

2. To the concept: TRPV4 activation leads to epithelial disintegrity and edema in ALI and this can be treated by antagonists. Perhaps formation of edema is necessary to neutralize an acid environment for instance or to dilute inflammatory mediators ...

The alveolar barrier consists of three layers. How does this fit into the concept?

3. It is possible to combine activation and successive inactivation in the same experiment?

4. Isolated buffer-perfused lung experiments could be valuable in further research.

Author Response

Manuscript Cells-3248583

Response to Reviewers

We thank both reviewers for their helpful comments. All changes according to the reviewers’ comments are marked in red in the revised manuscript.

Reviewer 2

Reviewer comment: The AT1 might be more instructive than ATI.

Author response: We appreciate the reviewer’s suggestion that Arabic numerals would be more distinct than the Roman numerals we employed when referencing ATI and ATII cells, and we have made the necessary replacements throughout the manuscript to reflect this.

Reviewer comment: To the concept: TRPV4 activation leads to epithelial disintegrity and edema in ALI and this can be treated by antagonists. Perhaps formation of edema is necessary to neutralize an acid environment for instance or to dilute inflammatory mediators ...

And

Reviewer comment: The alveolar barrier consists of three layers. How does this fit into the concept?

Author response: The reviewer raises two conceptual points: the first, that there may be a biological benefit to pulmonary edema as a way of protecting the lungs from a caustic or inflammatory environment. Although the consensus in the literature with regards to pulmonary edema is that it is mainly detrimental to the organism, we are now mentioning these additional two possible benefits on page 10 line 412-414.  

The second important point made by the reviewer is that the alveolar barrier includes the alveolar epithelial lining, the interstitium, and the microvascular endothelium. As TRPV4’s roles in endothelial barrier dysfunction have been extensively researched, we chose to narrow our focus to the relatively understudied alveolar epithelial layer. However, it could be the case that the connection between TRPV4 and ADAM10 that was observed in our AT1 cells also occurs in the microvascular endothelium.

We have modified our discussion to better address these points by elaborating on the potential role of TRPV4-ADAM10-E-cadherin signaling in the alveolar unit as a whole, and the importance of barrier permeability in the context of neutrophil transmigration (page 10, lines 425-437).

Reviewer comment: It is possible to combine activation and successive inactivation in the same experiment?

Author response: The experimental design suggested by the reviewer would be interesting as a preliminary assessment of the therapeutic potential for treating TRPV4-related lung edema with successive TRPV4 inactivation. However, we did not conduct these tests, as our primary aim with this manuscript was to identify and explore the signaling pathways underlying TRPV4-driven changes in alveolar epithelial barrier integrity. Indeed, two other TRPV4 inhibitors (GSK2220691 and GSK2337429A) applied after HCl exposure inhibited acute lung injury in vivo (Balakrishna et al. Am J Physiol Lung Cell Mol Physiol 307: L158–L172, 2014) and it would be interesting if GSK 219 is also able to act with similar efficiency in post-exposure experiments (see comment added in lines 381-385 of page 9). But we are also quite aware of the significant number of additional animals that would need to be sacrificed for the isolation of the AT1 cells required for these experiments. If such experiments are deemed necessary for this manuscript, we are prepared to perform them, and would ask for an extension of at least 6 weeks.

Reviewer comment: Isolated buffer-perfused lung experiments could be valuable in further research.

Author response: We agree with the reviewer that translation of these in vitro findings to an ex vivo system, such as the isolated, ventilated and perfused murine lung model, would be an excellent avenue to explore, which is now mentioned together with a reference describing this method on page 11, line 446.